# Protocol for the 'Supporting Young Cancer Survivors who Smoke' study (PRISM): Informing the development of a smoking cessation intervention for childhood, adolescent and young adult cancer survivors in England

Morven C. Brown [ID][1,2]*, Vera Araújo-Soares[3], Roderick Skinner [ID][2,4], Jamie Brown [ID][5], Adam W. Glaser[6,7], Helena Hanratty[8], Martin G. McCabe[9,10], Ana-Ecaterina Amariutei[11], Sabrina Mauri[11], Linda Sharp[1,2]

1 Population Health Sciences Institute, Newcastle University, Newcastle upon Tyne, United Kingdom,
2 Newcastle University Centre for Cancer, Newcastle University, Newcastle upon Tyne, United Kingdom,
3 Centre for Preventive Medicine and Digital Health (CPD), Division for Prevention of Cardiovascular and Metabolic Disease, Medical Faculty Mannheim, Heidelberg University, Heidelberg, Germany, 4 Department of Paediatric and Adolescent Haematology and Oncology, Great North Children's Hospital, Newcastle upon Tyne Hospitals NHS Foundation Trust, Newcastle upon Tyne, United Kingdom, 5 Department of Behavioural Science and Health, University College London, London, United Kingdom, 6 Department of Paediatric Oncology, Leeds Children's Hospital, Leeds Teaching Hospitals NHS Trust, Leeds, United Kingdom, 7 Leeds Institute of Medical Research, University of Leeds, Leeds, United Kingdom, 8 Sheffield Teaching Hospitals NHS Foundation Trust, Sheffield, United Kingdom, 9 Division of Cancer Sciences, The University of Manchester, Manchester, United Kingdom, 10 The Christie NHS Foundation Trust, Manchester, United Kingdom, 11 Patient and Public Representatives for the Study, United Kingdom

* morven.brown@newcastle.ac.uk

## Abstract

### Background

Childhood, adolescent and young adult (CAYA) cancer survivors are vulnerable to adverse late-effects. For CAYA cancer survivors, tobacco smoking is the most important preventable cause of ill-health and early death. Yet, effective strategies to support smoking cessation in this group are lacking. The PRISM study aims to undertake multi-method formative research to explore the need for, and if appropriate, inform the future development of an evidence-based and theory-informed tobacco smoking cessation intervention for CAYA cancer survivors.

### Materials and methods

PRISM involves three phases of: 1) an environmental scan using multiple strategies to identify and examine a) smoking cessation interventions for CAYA cancer survivors that are published in the international literature and b) current smoking cessation services in England that may be available to, or tailorable to, CAYA cancer survivors; 2) a qualitative study involving semi-structured interviews with CAYA cancer survivors (aged 16–29 years and

**Data Availability Statement:** This a protocol paper and there is no associated data. For future publications resulting from this study, data will be assigned a persistent identifier (DOI) that will allow the data to be discoverable but not openly accessible. The DOI will be included in any data access statement in publications. The corresponding author can be contacted for the data availability of future publications.

**Funding:** This article presents independent research funded by the National Institute for Health Research (NIHR) under the Research for Patient Benefit programme [NIHR202768]. The project was subjected to an external peer review as part of the application process. The views expressed in this article are those of the authors and not necessarily those of the NHS, the NIHR or the Department of Health. The funders had no role in study design, data collection and analysis, decision to publish, or preparation of the manuscript.

**Competing interests:** The authors have declared that no competing interests exist.

who are current or recent ex-smokers and/or current vapers) to explore their views and experiences of smoking, smoking cessation and vaping; and 3) stakeholder workshops with survivors, healthcare professionals and other stakeholders to consider the potential for a smoking cessation intervention for CAYA cancer survivors and what such an intervention would need to target and change. Findings will be disseminated to patient groups, healthcare professionals and researchers, through conference presentations, journal papers, plain English summaries and social media.

## Discussion

PRISM will explore current delivery of, perceived need for, and barriers and facilitators to, smoking cessation advice and support to CAYA cancer survivors from the perspective of both survivors and healthcare professionals. A key strength of PRISM is the user involvement throughout the study and the additional exploration of survivors' views on vaping, a behaviour which often co-occurs with smoking. PRISM is the first step in the development of a person-centred, evidence- and theory-based smoking cessation intervention for CAYA cancer survivors who smoke, which if effective, will reduce morbidity and mortality in the CAYA cancer survivor population.

## Introduction

Childhood, adolescent and young adult (CAYA) cancer survivors are a growing population. Approximately 4000 young people aged 0–24 years are diagnosed with cancer in the United Kingdom (UK) annually [1, 2]. Due to treatment advances, over 80% of those diagnosed will achieve long-term cure [1, 2]. However, late-effects of some cancer treatments—in particular, pulmonary and cardiac toxicities—can leave survivors vulnerable to chronic health conditions.

For CAYA cancer survivors, as with the general population, smoking tobacco is the most important health behaviour associated with ill-health and early death [3]. Smoking is particularly risky for this group because their health is already compromised by the cancer and its treatment. Smoking exacerbates survivors' risks of cardiovascular and respiratory diseases, and increases risk of a subsequent cancer, making these the most common causes of morbidity and mortality in this population [4, 5].

Children diagnosed with cancer have 3.6 to 6.4-fold increased risk of developing another cancer in later life compared to those in the general population [6]. Using both non-cancer controls and population data, those diagnosed in adolescence or young adulthood have been found to have a 2 to 3-fold increased risk of developing smoking-related cancers (e.g., lung) [6–8]; this rises to a 5-fold excess risk among some subgroups (e.g., survivors of Hodgkin lymphoma) [7]. British CAYA survivors diagnosed up to 19 years of age have 3–4 times excess risk of cardiac mortality than the general population [9, 10]. In addition, by the age of 40, half of UK survivors have been admitted to hospital for a respiratory condition [11], and the risk of respiratory mortality is raised 2-fold in those diagnosed with cancer in adolescence or young adulthood, and 7-fold in those diagnosed as children compared to what would be expected in the general population [12]. Despite these risks, a substantial proportion of CAYA survivors in the UK and United States (US) smoke tobacco: surveys of self-reported smoking cite figures of 14–35% [13–16]. However, it is worth nothing that these may be underestimates as in clinical groups, where smoking is especially stigmatized (e.g., pregnant women, patients with cardiac disease, cancer patients), patients may be particularly reticent about disclosing their tobacco use [17, 18].

Clinical practice guidelines for the follow-up care of CAYA cancer survivors from the UK and US, as well as recent harmonized international guidelines [19–22], state that survivors should be advised on both tobacco smoking and smoking cessation. However, a large proportion of CAYA survivors (including those who smoke) report not receiving such advice [23]. A UK survey of 95 healthcare professionals (HCPs) caring for these survivors found that only 50% of physicians and 36% of nurses reported providing smoking advice to most patients, with many stating they felt they were not the right person to do so [24]. This echoes findings from adult oncology in the UK, which shows that HCPs can feel uneasy discussing smoking with survivors, and lack awareness and knowledge of smoking cessation services [25]. Moreover, it is not known what—and how—smoking cessation support should be provided to CAYA cancer survivors to promote successful quitting, and whether existing services could support this (either in their current format, or if appropriately adapted). However, it also needs to be highlighted that although there is much evidence to support effective strategies for smoking cessation in adults, and the provision of cessation services, there is still limited evidence for how to effectively help young people in general to quit smoking.

There is limited empirical research on the views and attitudes of CAYA cancer survivors regarding smoking [23, 26]. Although cancer diagnosis, treatment and survivorship are often thought to offer 'teachable moments' for a lifestyle change such as quitting smoking [27], evidence suggests that survivors of adult cancers experience barriers to smoking cessation that are specific to their illness experience (e.g., smoking to help with cancer-related stress and to maintain personal control after their diagnosis [25, 27]), and which may affect the uptake, and effectiveness, of current smoking cessation services and support [28]. Whether this also holds in survivors of CAYA cancer is unclear and fundamental evidence is lacking on why CAYA cancer survivors' smoke, what affects whether they want to quit and what helps and hinders successful quitting.

In exploring CAYA survivors' views of smoking, it would be pertinent to also explore their views of e-cigarette use (also known as vaping). In England, around a quarter of young people aged 16–24 use e-cigarettes, whilst 14–20% smoke [29, 30]; evidence suggests that around half of those who vape, also smoke tobacco (i.e., are dual users) [29]. Dual users may find it harder to quit smoking and may not view themselves as 'smokers' [31], thus they may not perceive a study, or indeed an intervention, for smoking cessation to be relevant to them.

There is an urgent need to develop effective strategies to support smoking cessation among CAYA cancer survivors [3]. However, few interventions exist and of those that have been published [16, 32–34], none appear to have involved: (i) a systematic framework of intervention development; (ii) a thorough understanding of smoking behaviours in CAYA survivors; (iii) user involvement; or (iv) application of appropriate theory. These are prerequisites for developing successful interventions [35–38], suggesting considerable groundwork is needed to inform the development of effective smoking cessation interventions for CAYA cancer survivors.

The PRISM study will begin to address tobacco smoking in CAYA cancer survivors by undertaking formative research in order to provide a foundation for the future development of a smoking cessation intervention targeted to this population, with the ultimate goal of reducing smoking-related morbidity and mortality.

## Materials and methods

### Aim

PRISM aims to inform the future development of a person-centred, evidence-based and theoretically-informed tobacco smoking cessation intervention which can be tailored, as required, to the needs of CAYA cancer survivors.

## Objectives

1. a) To identify and describe the features of tobacco smoking cessation *interventions* for CAYA cancer survivors which have been published internationally; b) To identify and describe the features of tobacco smoking cessation *services* which currently exist for i) adolescents and/or young people with a medical diagnosis (cancer or another diagnosis) or ii) people of any age with cancer/cancer survivors in England;

2. To identify and explore perceived influences on tobacco smoking and vaping behaviour among CAYA cancer survivors;

3. To identify and explore perceived barriers to, and facilitators for, tobacco smoking cessation among CAYA cancer survivors;

4. To explore CAYA cancer survivors' views and experiences of tobacco smoking cessation advice and services, and what may help or hinder them to engage with such advice/services;

5. To engage with survivors, healthcare professionals and other key stakeholders to develop a preliminary logic model of the problem and identify areas for future development of a tobacco smoking cessation intervention for CAYA cancer survivors.

## Overall study design

PRISM will address shortcomings of previous research by following established guidance to generate evidence to inform intervention development [35–38]. It focuses on the planning domain of intervention development—that is, it will seek to understand the problem being addressed, agree the aims and goals of the future intervention, identify possible ways of addressing the problem, and consider real-world issues which may affect future implementation [39].

Phase 1 will involve an environmental scan where we will identify smoking cessation interventions for CAYA cancer survivors that have been published internationally. We will also identify existing smoking cessation services in England that may be relevant for CAYA cancer survivors (objective 1). Phase 2 will involve semi-structured interviews with 25–30 CAYA cancer survivors who are current or recent ex-smokers and/or current vapers (objectives 2–4). Evidence gathered from these phases will feed into Phase 3 (objective 5), which will involve workshops with key stakeholders and development of a preliminary logic model of the problem which will begin to define what an intervention will need to target and change, and its expected outcomes [37, 40].

The methods for each phase are presented below.

**Phase 1: Environmental scan.** Environmental scans seek, gather and interpret data from a wide range of sources, enabling assessment of the current state of healthcare services [41]. We will utilise multiple strategies to identify and characterise existing smoking interventions and cessation services which are either currently available to, or which could be adapted to the needs (retrofitted), of CAYA cancer survivors in England. These strategies will encompass: 1) a scoping search of published literature; 2) a comprehensive search of grey literature and 3) consultation with key informants and stakeholders.

*Scoping search of published literature.* We will search bibliographic databases (MEDLINE, Embase, PsycINFO, CINAHL, Scopus) to identify smoking cessation interventions with CAYA cancer survivors in the published literature. Search strategies will be informed by PICO search strategy tool [42]. Search terms will relate to CAYA cancer survivors, interventions and smoking cessation and a combination of subject headings and key words and will be adapted for each database. Searches will be limited to the English language.

*Comprehensive search of grey literature.* To rigorously identify grey information, we will search: 1) clinical trial registries (ISRCTN registry, ClinicalTrials.gov) and a grey literature database (OpenGrey); 2) results from a popular Internet search engine (Google.co.uk); 3) websites of relevant organisations (e.g., National Health Service (NHS); and 4) App stores (Google Play and Apple App Store) [43, 44].

A Medical Sciences Librarian will advise on a customised search strategy for each source. Search strategies will be based on two components–the intervention/services of interest (e.g., smoking cessation) and population of interest (e.g., young people, cancer patients/survivors).

*1. Grey literature and trial database.*   In trial databases, advanced search functions will be utilised to identify smoking cessation trials. Separate searches will be run with restrictions for participant age (child) and condition/disease (cancer). Searches will include all trial statuses (e.g., ongoing, completed, suspended) and will be limited by country (England/UK, as database allows). The OpenGrey database will also be searched.

*2. Google.*   Multiple combinations of terms for the two search components will be run using Google Advanced Search. Searches will be restricted to the UK and language (English) and to the first 100 results of each search. Results will be archived by copying results into a Microsoft Word document, retaining the page titles, site links and brief description of the page to enable review for eligibility.

*3. Websites.*   Using Google Advanced Search, searches will use a combination of key terms and will be restricted by domain name in order to search the content of specific websites (e.g., cancerresearchuk.org) and language (English). The first 50 results of each search will be archived for review.

*4. App stores.*   Google Play (via play.google.com) and the Apple App store (via an Iphone13) will be searched using lay language keywords for smoking cessation (e.g., stop smoking, quit smoking), limiting to the first 100 results. App stores do not enable results to be exported, therefore eligibility will be determined by one researcher based on the app's marketing description. Any apps deemed potentially relevant will be downloaded.

*Consultation with key informants and stakeholders.* **Consultation with key informants.** To identify services we will consult with key informants (e.g., smoking cessation services, Cancer Alliances, relevant HCPs in primary and secondary/tertiary care, local public health departments, smoking cessation researchers). We will use a snowball approach by asking contacts if they are aware of other relevant organisations or individuals.

**Healthcare professional survey.**   We will disseminate a brief online survey via professional associations for HCPs involved in the care of CAYA cancer survivors (Children's Cancer and Leukaemia Group; Teenagers and Young Adults with Cancer; Teenage Cancer Trust) and social media. The survey will seek information on which (if any) smoking cessation services HCPs refer patients/survivors to. Additional questions will investigate current practices of HCPs with regards to offering advice and support about smoking/smoking cessation to survivors, and the perceived influences on these behaviours. These questions will be informed by the Theoretical Domains Framework (TDF) [45, 46], a widely used theoretical approach for identifying determinants of behaviour.

*Identifying relevant interventions and services.* Eligible studies identified via the scoping search of published literature must report on a smoking cessation intervention for CAYA cancer survivors in any geographical location. To be eligible for inclusion in the grey literature scan, services, interventions and apps must state that they are targeted towards either 1) adolescents and/or young people with a medical diagnosis (cancer or another diagnosis) or 2) people of any age with cancer/cancer survivors. These interventions and services must be available to individuals residing in England, and any apps must be in English. We envisage that few, if any, services will be aimed specifically to CAYA cancer survivors; conducting a wider search

will therefore identify services which may provide a basis for a future smoking cessation intervention for CAYA survivors.

*Data extraction and analysis*. Information will be extracted on the name of service/intervention; target population (who is eligible to access the service); any tailoring; how it is accessed; location and/or type or format; aim and description of what it entails; who provides or delivers it; how long individuals are enrolled in the service and how frequently they are asked to attend (if applicable); how many individuals access it annually; cost (if applicable); and whether it has been evaluated [47]. Excel will be used to build a database of extracted information to enable synthesis. Narrative synthesis will be undertaken [48].

**Phase 2: Qualitative study.** *Study setting and participants*. Phase 2 will involve semi-structured interviews with CAYA cancer survivors who are current or recent tobacco smokers and/or current vapers (Table 1 shows eligibility/exclusion criteria). Survivors will be primarily recruited via four clinical sites (Newcastle upon Tyne Hospitals NHS Foundation Trust; Leeds Teaching Hospitals NHS Trust; Sheffield Teaching Hospitals NHS Foundation Trust; and The Christie NHS Foundation Trust) which provide care for CAYA cancer survivors. To reach a wider range of survivors we will also promote the study via the social media pages of the research team and, with permission, the pages of local charities (e.g., Children's Cancer North). Phase 2 received approval from West Midlands–South Birmingham Research Ethics Committee (REC reference: 22/WM/0102).

*Recruitment*. Identification and screening of CAYA cancer survivors began in January 2023, with recruitment expected to end 01 July 2024. Young people's disclosure of tobacco smoking can be influenced by the setting they are in, the perceived disapproval of others and anonymity [49], and indeed whether they view themselves as a smoker or not. Therefore, we will adopt a range of approaches for recruitment. Maximum variation purposive sampling (with strata including age, gender, ethnicity, socio-economic status, diagnosis, treatment) will be used to ensure elicitation of varied views and experiences.

The primary recruitment route will be via CAYA cancer survivor follow-up services in the Trusts. HCPs will identify potentially eligible survivors through medical records and at clinic, and inform them about the study by letter, telephone or verbally face-to-face. All those approached will receive a participant information sheet. Those interested in participating will either contact the study's researcher directly or give permission for their details to be provided to the researcher. Where possible, the researcher will attend clinics to be able to speak to eligible patients about the study. We have successfully used these approaches to recruit CAYA survivors to several studies [50, 51]. In describing the study to survivors (both verbally and via the participant information sheet) neutral, non-judgmental language will be used. It will be made clear that it is not assumed that they wish to stop smoking or vaping, nor does the study involve efforts to persuade them to do so; we simply want to hear their views on smoking and vaping.

**Table 1. Childhood, adolescent and young adult cancer survivors' inclusion and exclusion criteria.**

| Inclusion criteria | Exclusion criteria |
|---|---|
| 1) Were diagnosed with cancer or a brain tumour aged 0–24 years<br>2) Are currently aged 16–29 years<br>3) Are cancer free and at least 12 months from the end of treatment<br>4) Have the ability to speak and understand English<br>5) Are a regular current tobacco smoker or an ex-smoker who stopped smoking tobacco within the last 12 months and/or are a current user of e-cigarettes | 1) Only smoke tobacco when mixed with cannabis (and do not use e-cigarettes)<br>2) Have a cognitive impairment which would significantly impact their understanding of the study or their ability to give informed consent<br>3) Are unfit to take part due to issues such as the presence of serious psychological problems which may mean they would find taking part distressing (as deemed by the oncologist/nurse specialist) |

These targeted methods rely on the HCPs identifying CAYA survivors whom they know, or suspect, to smoke tobacco or vape. If required, this approach will be supplemented by Trusts distributing a brief screening questionnaire to survivors who fulfil inclusion criteria 1–4 (Table 1). Those who declare themselves to be current smokers, recent ex-smokers and/or current vapers will be invited to provide their contact details if they are interested in receiving information about the study. Survivors will return the questionnaire in a sealed envelope to the researcher to ensure confidentiality.

For recruitment via social media, posts promoting the study will ask interested survivors to contact the researcher who will screen them against the inclusion criteria and send them a participant information sheet.

*Procedures/data collection and analysis*. CAYA cancer survivors wishing to take part will be able to choose their preferred mode of interview—telephone, online (e.g., Zoom) or face-to-face at a location of their choice. Offering multiple options can increase research participation of more marginalised groups, or those traditionally less likely to participate [52]. Prior to interview, participants will provide informed consent. Written consent will be obtained from those interviewed in person. For interviews conducted remotely, verbal consent will be audio-recorded. The researcher will read out each point of the consent form and will ask the participant to indicate their agreement or otherwise. The researcher will then complete a form on the participant's behalf. Interviews will be guided by a topic guide, which will be used flexibly (available in S1 File). Interviews will adopt a non-judgemental attitude and will explore: cancer beliefs; awareness of late-effects; smoking and vaping-related health beliefs/risk perceptions; views and experiences of smoking cessation advice/resources/services; quit attempts; perceived influences on their smoking and/or vaping behaviour.

Elements of the interview will be informed by the TDF to: help support comprehensive assessment of behavioural determinants; enable subsequent identification of the most relevant theory; and identify what needs to be targeted by strategies to bring about behaviour change [45]. As recommended [53], the TDF will be used flexibly to support identification of key determinants of (i) tobacco smoking and vaping and (ii) successful smoking cessation. Interviews are expected to last approximately 60–90 minutes. All interviewees will be offered a £20 shopping voucher to thank them for their time, as well as reimbursement of any travel expenses.

Interviews will be audio-recorded and transcribed verbatim. Inductive reflexive analysis will occur concurrently with data collection to ensure any new issues raised are explored in subsequent interviews [54]. Two team members will code preliminary interviews, discuss and agree codes and themes. These codes will be applied to the remaining interviews, incorporating any new codes and themes as they are identified. In a second, deductive, phase of analysis, codes which relate to the influences on smoking/smoking cessation (and vaping) will be mapped onto the TDF, distinguishing between those factors which can, and cannot, be modified. Coding and analysis will be facilitated by QSR International's NVivo software (Release 1.6, 2022).

*Sample size and data saturation*. Recruitment will continue until reasonable data saturation has been achieved [55]. Determination of data saturation will be primarily based on theoretical sufficiency (conceptual depth) regarding survivors' perceived influences on smoking behaviour. In addition, we will ensure no new themes have been identified in the last three interviews as regards other objectives [56]. Based on our past research [50, 51], and recommendations for sample size for semi-structured interviews [57], we anticipate 25–30 interviews will be required.

**Phase 3: Stakeholder workshops.** *Study setting and participants*. Phase 3 will involve working with CAYA cancer survivors, HCPs (e.g., oncologists, nurses, general practitioners)

and other stakeholders (e.g., public health, smoking cessation service providers) to begin to identify what a smoking cessation intervention for CAYA cancer survivors will need to target.

We will hold two (parallel) workshops, one for survivors, and the other for HCPs (oncologists, nurses, general practitioners) and other stakeholders (e.g., public health, service providers). If required, and if time permits, additional workshops may be held.

*Recruitment to survivor workshop.* CAYA survivors who participate in Phase 2 will be asked to register their interest in workshop attendance. Clinical co-applicants/collaborators will advertise this Patient and Public Involvement (PPI) opportunity via their service and appropriate links at their Trusts (e.g., support groups). We will also invite members of our network of CAYA cancer survivors interested in PPI in Northern England to participate. For greater reach, the opportunity will be advertised via cancer charities and organisations (e.g., Children's Cancer North), support groups and social media.

*Recruitment to professional workshop.* Stakeholders (e.g., public health, smoking cessation service providers) will be identified via Phase 1. HCPs (e.g., oncologists, nurses, general practitioners) who care for CAYA cancer survivors will be identified via the contacts of the clinical co-applicants and advertisement via professional associations (e.g., Children's Cancer and Leukaemia Group) and social media.

*Procedures/data collection and analysis.* If the workloads of HCPs permit, workshops will take place face-to-face at an easily accessible location; if this is not possible, remote workshops will be held using a video conferencing platform that is accessible to workshop participants (e.g., Zoom for survivors, MS Teams for HCPs). These workshops will draw on co-design approaches [58], in that the research team and stakeholders will work together during them to co-create an understanding of the problem and the potential for a solution.

At each workshop, the Phase 1 and 2 findings will be presented (in the form of evidence statements) [58], and initial ideas for what would be useful, and what is needed, to promote smoking cessation among CAYA cancer survivors will be explored. The survivor workshop will explore issues around the acceptability of any intervention to support smoking cessation (e.g., perceived need for intervention, potential stigma of using intervention), factors which may encourage or prevent CAYA survivors engaging in an intervention (e.g., referral by HCP, low readiness to quit), and initial views on intervention delivery (e.g., where, how and by whom). Attendees at the survivor workshop will be offered a £75 honorarium [59]. The workshop with HCPs and wider stakeholders will begin to consider issues around the feasibility of health service/HCP involvement in any intervention and potential contextual factors that might affect an intervention. Based on these workshops, the logic model of the problem will be refined and potential areas for future intervention development identified.

**Dissemination of findings.** To reach academic and clinical audiences, dissemination will be via journal publications and conferences. We will work with our PPI co-applicants on dissemination methods to reach CAYA cancer survivors. We will produce a plain English summary of the project results for dissemination through HCP and survivor networks and organisations (e.g., Children's Cancer and Leukaemia Group).

**Patient and public involvement.** PPI input shaped the study. A brief survey about smoking/smoking cessation was posted in two Facebook groups for CAYA cancer survivors. Discussions were held with members of *Perspectives*, an adult cancer PPI group. Members of the *Young Person's Advisory Group North-East* provided input on the content of participant information sheets, recruitment methods and the appropriateness and conduct of interviews. PPI co-applicants (AA and SM) will advise and assist with strategies and patient-facing materials for recruitment to Phase 2 and 3. They will help to shape project direction and decision making, and be invited to contribute to the interpretation and dissemination of study findings. We will use the GRIPP2 short form checklist to report PPI involvement in our research in publications [60].

## Discussion

The increased risk of smoking-related health conditions among CAYA cancer survivors indicates the need for smoking cessation interventions, but high-quality, evidence-based and theoretically-informed interventions are currently lacking. There is also a lack of research exploring CAYA survivors' perceived needs for smoking cessation advice and support, and perceived barriers and facilitators to smoking cessation. To our knowledge, there are also no studies which explore the views of CAYA cancer survivors who vape, a behaviour which may have important implications for smoking cessation efforts. PRISM will generate this information to underpin future development of tobacco smoking cessation interventions for this group.

Evidence suggests that cancer professionals lack awareness of what smoking cessation services are available [25]. By mapping the smoking cessation landscape in England, PRISM will both identify any interventions/services which could potentially be retrofitted to CAYA cancer survivors (thus enabling agile intervention development) [35], and provide information that may help professionals in the short-term to implement the guidelines advising CAYA survivors on smoking cessation [19–21].

Tobacco smoking often co-occurs with other behaviours such as vaping e-cigarettes and smoking cannabis [61, 62]. These are related but distinct behaviours, all of which have very different drivers [61–64]. In the UK, e-cigarettes are tightly regulated and recommended as a smoking cessation aid for those aged 18 and above and while 16% of young people report using cannabis [65], it remains to be illegal. PRISM focuses specifically on smoking tobacco as CAYA cancer survivors have increased risk of tobacco-related diseases but due to the complexities that may be caused by dual use of e-cigarettes and tobacco cigarettes (e.g., dual users not identifying as smokers), we also include CAYA survivors who vape in this study. Contrary to the public health recommendations of other countries (e.g., Australia, the US), vaping is promoted as a much safer alternative to smoking in the UK. However, little is known about the possible long-term health risks of vaping, and emerging evidence suggests links to increased risk of respiratory and cardiovascular diseases [66], which should be of particular concern to CAYA survivors. CAYA cancer survivors who smoke cannabis will also be eligible for Phase 2 provided they also smoke tobacco separately (and/or vape), thus the study may also provide some (albeit limited) information about cannabis smoking in this population.

Adolescence is a critical period for smoking initiation, therefore, addressing tobacco use in this patient group will likely benefit from efforts to both prevent uptake in survivors and also support smoking cessation for those who already smoke. By speaking to survivors who smoke and vape we hope to be able to explore their views on the initiation of these behaviours (e.g., when and why they started), by doing so, this may provide information useful to the consideration of smoking prevention in this group.

PRISM is the first step in the development of a person-centred, evidence- and theory-based smoking cessation intervention aimed at CAYA cancer survivors who smoke tobacco. This intervention will seek to support smoking cessation, in order to reduce long-term morbidity and mortality. Therefore, in the long-term the project has considerable potential to yield significant benefits for the CAYA cancer survivor population. In addition, if, in the long-term, it results in an effective tobacco smoking cessation intervention for CAYA cancer survivors, it could yield significant cost savings for the NHS, by preventing admissions of respiratory and cardiac conditions and second cancers.

## Supporting information

**S1 File.**
(PDF)

## Author Contributions

**Conceptualization:** Morven C. Brown, Vera Araújo-Soares, Roderick Skinner, Linda Sharp.

**Funding acquisition:** Morven C. Brown, Vera Araújo-Soares, Roderick Skinner, Jamie Brown, Adam W. Glaser, Helena Hanratty, Martin G. McCabe, Ana-Ecaterina Amariutei, Sabrina Mauri, Linda Sharp.

**Methodology:** Morven C. Brown, Vera Araújo-Soares, Roderick Skinner, Jamie Brown, Adam W. Glaser, Helena Hanratty, Martin G. McCabe, Linda Sharp.

**Project administration:** Morven C. Brown.

**Supervision:** Vera Araújo-Soares, Roderick Skinner, Linda Sharp.

**Visualization:** Jamie Brown, Ana-Ecaterina Amariutei, Sabrina Mauri.

**Writing – original draft:** Morven C. Brown.

**Writing – review & editing:** Morven C. Brown, Vera Araújo-Soares, Roderick Skinner, Jamie Brown, Adam W. Glaser, Helena Hanratty, Martin G. McCabe, Ana-Ecaterina Amariutei, Sabrina Mauri, Linda Sharp.

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
