## [Decision Letter · Decision Letter 0]

13 Dec 2023

PONE-D-23-33074Protocol for the Supporting Young Cancer Survivors who Smoke Study (PRISM): Informing the development of a smoking cessation intervention for childhood, adolescent and young adult cancer survivorsPLOS ONE

Dear Dr. Brown,

Thank you for submitting your manuscript to PLOS ONE. After careful consideration, we feel that it has merit but does not fully meet PLOS ONE’s publication criteria as it currently stands. Therefore, we invite you to submit a revised version of the manuscript that addresses the points raised during the review process.

**This manuscript addresses an incredibly important area for clinical care and public health. Please see my specific comments below in the section "**Additional Editor Comments".==============================

We look forward to receiving your revised manuscript.

Kind regards,

Jesse T. Kaye, PhD

Academic Editor

PLOS ONE

Journal Requirements:

"This article presents independent research funded by the National Institute for Health Research (NIHR) under the Research for Patient Benefit programme [NIHR202768]. The project was subjected to an external peer review as part of the application process. The views expressed in this article are those of the authors and not necessarily those of the NHS, the NIHR or the Department of Health."

Additional Editor Comments:

I thank Dr. Brown and colleagues for this report of a protocol for a multi-phased study of that aims to inform smoking cessation interventions for survivors of cancer during young adulthood and earlier. This represents an incredibly important topic as much is still unknown regarding the best approaches to engage this population in tobacco treatment and the most effective intervention components. The evidence is clear regarding the detrimental health effects of smoking after a cancer diagnosis and treatment. However, we still don't know the best way to treat smoking in this population - and survivors of cancer tend to be under-treated due to a variety of individual, clinician, and system level factors. It is not clear if treatment tailoring is required or beneficial for this population, though it warrants research into the key stakeholders beliefs in this area. I applaud the patient-centered approach to this protocol design. The two reviewers provide thoughtful comments that I encourage the authors to consider in their revision. Of course, all comments may not be addressable given that this study is already partially underway; but those aspects that can be addressed I would encourage thoughtful integration.

Reviewer 2 provides very comprehensive suggestions that I believe will improve both the manuscript and the execution of the overall project. As PLOS One has an international readership, highlighting some of the unique features of the UK healthcare system and tobacco control approach (including re e-cigarettes) is appropriate. The reviewer also highlights a number of valuable resources from the US (e.g., Surgeon General Report, NCI Monograph 23) that would likely have many relevant studies that are relevant to the UK tobacco treatment landscape as well. In addition to these resources, I would point the authors to one other website resource from the Cancer Center Cessation Initiative (C3I) in the US, supported by the National Cancer Institute (https://www.oncologytobaccotreatment.org/). This C3I Roadmap provides a plethora of resources to help cancer treatment centers to incorporate tobacco assessment and treatment into standard of care for patients with and survivors of cancer. The website is a work in progress that will be developed over the next year. I do not point out this resource to necessarily be cited or referenced in this PLOS One manuscript, but thought it may be a useful resource in the future as the PRISM study progresses.

Lastly, I would encourage to the authors to include the semi-structured interview guides as a supplement to the resubmission if they are prepared and ready to disseminate. I realize that they may very likely be adapted based on the information gleaned from early interviews, but would still be a valuable resource for readers to be available.

Reviewers' comments:

Reviewer's Responses to Questions

**Comments to the Author**

1. Does the manuscript provide a valid rationale for the proposed study, with clearly identified and justified research questions?

Reviewer #1: Yes

Reviewer #2: Yes

2. Is the protocol technically sound and planned in a manner that will lead to a meaningful outcome and allow testing the stated hypotheses?

Reviewer #1: Yes

Reviewer #2: Yes

3. Is the methodology feasible and described in sufficient detail to allow the work to be replicable?

Reviewer #1: Yes

Reviewer #2: Yes

4. Have the authors described where all data underlying the findings will be made available when the study is complete?

Reviewer #1: Yes

Reviewer #2: Yes

5. Is the manuscript presented in an intelligible fashion and written in standard English?

Reviewer #1: Yes

Reviewer #2: Yes

6. Review Comments to the Author

You may also provide optional suggestions and comments to authors that they might find helpful in planning their study.

Reviewer #1: Overall: A valuable and ambitious protocol.

Introduction: Well-written and compelling.

--Lines 57-64 should be re-written as I cannot understand the point being made in lines 58-62. I think the point is that this is a 'unique' population with unique barriers (stress) and facilitators (list one?) to healthy behavior that exist due to their cancer history. The first and last sentence are stating the same thing, would make the first sentence even more general and keep last sentence specific.

--Line 67: instead of "many" of those who vape. Could you list the percentage from the reference?

Materials and Methods: Well organized and thoughtful, comprehensive approach.

--line 85: Centred spelled incorrectly.

--Line 89 (Objective 1): you mention searching for interventions for young people with a "medical diagnosis (cancer or another diagnosis)" but then in the environmental scan (line 125), mention limiting the intervention search to those "with CAYACS". I think either option; limiting to CAYACS or keeping it broad "cancer or another diagnosis" is reasonable though the latter will certainly lead to a larger field of results. Would try to clarify objectives with search descriptions to make this consistent (you make this argument more clearly in lines 172-178).

Table 1: Personal preference but would take out the row bars between the criteria.

-Qualitative approach appears methodologically solid.

-Workshops in person appear unlikely so appreciate the backup option

Reviewer #2: This is an extremely well written and thoughtful protocol which addresses an important topic. The authors’ plan to engage survivors of childhood, adolescent, and young adult cancer themselves, as well as clinicians who may be involved in delivering cessation treatment to this population is a significant strength. Below are suggestions that the authors may find helpful to improve an already excellent manuscript.

General Comments:

I understand the rationale (saving space) for using the abbreviation CAYACS but note that it is not a standard abbreviation and may confuse some readers.

The proposed age range for study and intervention is quite broad (16-29 years). Will different populations within this broad age range need somewhat different approaches? Also, given that this age group encompasses the time period when people are most likely to initiate smoking, prevention of initiation and progress to established smoking should also be an important component of ensuring this population does use tobacco products. Although not the focus of this study, suggest noting that addressing tobacco use, through both prevention of onset and cessation after onset, in the population of childhood, adolescent and young adult cancer survivors is an area in need of additional research.

Research on cessation treatments for childhood, adolescent and young adult smokers should be put in the larger context of research studies on the general population of smokers. For example, I would suggest noting that while there are effective treatments to promote cessation among adult smokers (age 21 +) there is a limited evidence base to assist people younger than age 21 to quit. In 2020, the U.S. Preventive Services Task Force published a report which identified the need for additional studies “to identify effective interventions to help children and adolescents who use tobacco products to quit.” The report also provided specific suggestions for research studies. See: https://www.uspreventiveservicestaskforce.org/uspstf/recommendation/tobacco-and-nicotine-use-prevention-in-children-and-adolescents-primary-care-interventions#bootstrap-panel--9

The age range of 16-29 years includes the reproductive age for women. This topic is especially complex for survivors of childhood, adolescent and young adult cancer, but nonetheless may be a useful focus for discussion in the focus groups of patients and clinicians.

The title may want to make clear that the focus of the study is England. Related to this, the authors may consider ways to make the study generalizable to survivors of childhood, adolescent and young adult cancers living in other countries.

It is helpful to more clearly distinguish between studies, recommendations and guidelines that have been produced for England/U.K. and those produced for other countries. This is particularly important where studies address topics that are strongly influenced by policies and healthcare systems that differ across countries.

The study’s focus is on addressing cessation at the individual/clinical level is useful and appropriate, especially because this population typically has more contact with the health care system than same age peers. However, it is worth noting that population-level interventions (e.g., significant increases in tobacco taxes, smoke-free air laws, anti-tobacco counter marketing) have been shown to promote smoking cessation among the general population of smokers and are also likely to promote cessation in this specific population as well. See for example chapter 7 (Clinical-, System-, and Population-Level Strategies that Promote Smoking Cessation) of the 2020 U.S. Surgeon General’s Report, Smoking Cessation. Available here: https://www.cdc.gov/tobacco/sgr/2020-smoking-cessation/index.html

I would also draw the authors’ attention to this publication: NCI Tobacco Control Monograph 23, Treating Smoking in Cancer Patients: An Essential Component of Cancer Care. Smoking Cessation among childhood, adolescent and young adult cancer survivors is addressed on pages 92-95. Available here: https://cancercontrol.cancer.gov/brp/tcrb/monographs/monograph-23

As I’m sure the authors are aware, smoking cessation typically requires numerous attempts to quit, and tobacco dependence is often viewed as a “chronic condition” until complete and long-term cessation can be achieved. This emphasizes the need for continued vigilance when working with tobacco users, including those who are survivors of childhood, adolescent, and young adult cancers.

Specific Comments:

Lines 38-40 do not define the comparison/reference group. Suggest do so consistently.

Lines 44-46. Suggest explaining that while self-report is considered reliable in the general population, this is not true in cancer survivors and other populations (e.g., pregnant women) who feel significant stigma if they acknowledge tobacco use.

Line 65-70. Given the high prevalence of use of e-cigarettes (and dual use of e-cigarettes and conventional cigarettes) in this age group, exploring use of and views on e-cigarette use is important to consider.

Lines 78-80. As regards development of smoking cessation interventions targeted to this population, I would encourage the authors to consider both the content of the intervention and the proposed mode of delivery (e.g., in person during clinical visits, telephone quitlines, digital health modalities.)

Lines 108. Suggest providing the number of survivors who will be included in semi-structured interviews.

Line 123. As regards the scope of the literature search, suggest the authors consider where it makes sense to restrict to England/UK, and where it is reasonable to expand to other countries. While health care policy and delivery is typically country specific, the experience helping survivors to quit may well be relevant across countries.

Line 207. In addition to variables related to age, gender, diagnosis, and treatment, if efforts will be made to recruit a diverse sample of survivors (e.g., based on socio-economic status, race/ethnicity or other variables) suggest note this.

Line 243. I would suggest that the stakeholder interviews examine the survivors’ exposure to secondhand tobacco smoke, at work and in the home. (I am aware that smoking is banned in most workplaces in the UK, but exposure may still occur in these areas.) Exposure to secondhand smoke impedes quitting; conversely, workplace and home smoking bans facilitate quitting. As noted in this U.S. report: https://cancercontrol.cancer.gov/sites/default/files/2020-06/tus-cps_2014-15_summarydocument.pdf, “Smoking status is a strong predictor of whether a home smoking ban is reported, with current smokers far less likely to report a ban than former/never smokers.”

Line 328: Use of e-cigarettes has potential implications for smoking cessation efforts. However, it may also be worth noting that not enough is known about the short- and long-term health effects of e-cigarette use, specifically in the high-risk population of cancer survivors. As noted on line 338, e-cigarettes are recommended as a smoking cessation aid in the U.K.; the authors study may wish to consider whether this recommendation is appropriate for childhood, adolescent and young adult cancer survivors.

7. PLOS authors have the option to publish the peer review history of their article (what does this mean?). If published, this will include your full peer review and any attached files.

Reviewer #1: No

Reviewer #2: No

---

## [Author Response · Author response to Decision Letter 0]

6 Feb 2024

We thank the Editor and the reviewers for such positive and constructive comments. We hope we have addressed these satisfactorily and are particularly appreciative of the documents which have been suggested to us. 

Below we have addressed each comment. Where relevant we have indicated where the edit can be found in the clean manuscript.

Additional Editor Comments:

As PLOS One has an international readership, highlighting some of the unique features of the UK healthcare system and tobacco control approach (including re e-cigarettes) is appropriate. 

- Thank you. In response to this comment, and others from the reviewers, we have now added extra information regarding the situation in the UK and how this may differ from other public health approaches in other countries. This includes stating “e-cigarettes are tightly regulated and recommended as a smoking cessation aid for those aged 18 and above” (Discussion, line 368).

- Also “Contrary to the public health recommendations of other countries (e.g., Australia, the US), vaping is promoted as a much safer alternative to smoking in the UK.” (Discussion, line 373).

I would encourage to the authors to include the semi-structured interview guides as a supplement to the resubmission if they are prepared and ready to disseminate. 

- Yes, we are happy to provide this and it has been uploaded.

Reviewers' comments to the Author:

Reviewer #1: 

Lines 57-64 should be re-written as I cannot understand the point being made in lines 58-62. I think the point is that this is a 'unique' population with unique barriers (stress) and facilitators (list one?) to healthy behavior that exist due to their cancer history. The first and last sentence are stating the same thing, would make the first sentence even more general and keep last sentence specific.

- This paragraph has been edited and we hope it is clearer now (Introduction; line 72-80): There is limited empirical research on the views and attitudes of CAYA cancer survivors regarding smoking [23, 26]. Although cancer diagnosis, treatment and survivorship are often thought to offer ‘teachable moments’ for a lifestyle change such as quitting smoking [27], evidence suggests that survivors of adult cancers experience barriers to smoking cessation that are specific to their illness experience (e.g., smoking to help with cancer-related stress and to maintain personal control after their diagnosis [25, 27]), and which may affect the uptake, and effectiveness, of current smoking cessation services and support [28]. Whether this also holds in survivors of CAYA cancer is unclear and fundamental evidence is lacking on why CAYA cancer survivors’ smoke, what affects whether they want to quit and what helps and hinders successful quitting.

Line 67: instead of "many" of those who vape. Could you list the percentage from the reference?

- The sentence has been amended to (Introduction; line 82-84): In England, around a quarter of young people aged 16-24 use e-cigarettes, whilst 14-20% smoke [25, 26]; evidence suggests that around half of those who vape, also smoke tobacco (i.e., are dual users) [25].

Line 85: Centred spelled incorrectly.

- We have used the UK English spelling for centred (and also other words throughout the manuscript, such as behaviour, organisation). However, if the Editor requires US spellings we are happy to change throughout.

Line 89 (Objective 1): you mention searching for interventions for young people with a "medical diagnosis (cancer or another diagnosis)" but then in the environmental scan (line 125), mention limiting the intervention search to those "with CAYACS". I think either option; limiting to CAYACS or keeping it broad "cancer or another diagnosis" is reasonable though the latter will certainly lead to a larger field of results. Would try to clarify objectives with search descriptions to make this consistent (you make this argument more clearly in lines 172-178).

- Thank you for highlighting this. We have amended the text to distinguish between a) our search for smoking interventions for CAYA cancer survivors that have been published in the international literature and b) our search for smoking cessation services in that are available in England, and may be adaptable to CAYA cancer survivors. The text for objective one now is as follows (Materials and methods; line 105-109: 1. a) To identify and describe the features of tobacco smoking cessation interventions for CAYA cancer survivors which have been published internationally; b) To identify and describe the features of tobacco smoking cessation services which currently exist for i) adolescents and/or young people with a medical diagnosis (cancer or another diagnosis) or ii) people of any age with cancer/cancer survivors in England;

Table 1: Personal preference but would take out the row bars between the criteria.

- The borders between rows have now been deleted.

Reviewer #2: 

I understand the rationale (saving space) for using the abbreviation CAYACS but note that it is not a standard abbreviation and may confuse some readers.

- Yes, we entirely agree with the reviewer. This acronym was used to reduce the word count. We are more than happy to change this to childhood, adolescent and young adult (CAYA) cancer survivors and use CAYA cancer survivors or CAYA survivors throughout the rest of the paper. 

The proposed age range for study and intervention is quite broad (16-29 years). Will different populations within this broad age range need somewhat different approaches?

- The aim of this project is to undertake formative research which will generate evidence of the factors which either help or hinder survivors attempts to quit, and which will enable us to have a better understanding of the support which may be needed by survivors. During analysis, we may be able to identify if different approaches are needed and if these appear to be associated with particular groups/characteristics (e.g., younger vs older survivors; female vs male survivors).

- The phase 3 workshops aim to discuss issues and generate ideas concerning smoking cessation interventions for CAYA survivors, such as: is a there a need to develop a smoking cessation intervention for this patient group? How acceptable or feasible would a smoking cessation intervention be in this patient group? What does the intervention need to target in order to help survivors to increase their chances of successfully quitting? Therefore, we expect issues such as the one raised above, to be discussed at this point. 

- This project does not aim develop an intervention, but to explore the need for one, and begin to generate which will be used to inform future development efforts.

- We have slightly edited our aim to make it clearer that this intervention will be designed so that it can be tailored to the needs of CAYA survivors: PRISM aims to inform the future development of a person-centred, evidence-based and theoretically-informed tobacco smoking cessation intervention which can be tailored, as required, to the needs of CAYA cancer survivors. 

Also, given that this age group encompasses the time period when people are most likely to initiate smoking, prevention of initiation and progress to established smoking should also be an important component of ensuring this population does use tobacco products. Although not the focus of this study, suggest noting that addressing tobacco use, through both prevention of onset and cessation after onset, in the population of childhood, adolescent and young adult cancer survivors is an area in need of additional research.

Yes we agree that prevention of smoking in CAYA survivors is also an area of importance. We have added the following sentence in the discussion (Discussion; line 380-385): Adolescence is a critical period for smoking initiation, therefore, addressing tobacco use in this patient group will likely benefit from efforts to both prevent uptake in survivors and also support smoking cessation for those who already smoke. By speaking to survivors who smoke and vape we hope to be able to explore their views on the initiation of these behaviours (e.g., when and why they started), by doing so, this may provide information useful to the consideration of smoking prevention in this group.

Research on cessation treatments for childhood, adolescent and young adult smokers should be put in the larger context of research studies on the general population of smokers. For example, I would suggest noting that while there are effective treatments to promote cessation among adult smokers (age 21 +) there is a limited evidence base to assist people younger than age 21 to quit. 

- Thank you for highlighting the U.S. Preventive Services Task Force document to us. To address the above point we have added the following to the introduction (Introduction; line 69-71)): However, it also needs to be highlighted that although there is much evidence to support effective strategies for smoking cessation in adults, and the provision of smoking cessation services, there is still limited evidence for how to effectively help young people in general to quit smoking.

The age range of 16-29 years includes the reproductive age for women. This topic is especially complex for survivors of childhood, adolescent and young adult cancer, but nonetheless may be a useful focus for discussion in the focus groups of patients and clinicians.

- Thank you for this suggestion. We expect that this issue may be raised in the interviews with young women who smoke. However, we agree this is something we should also take forward to the discussion groups in Phase 2.

The title may want to make clear that the focus of the study is England. Related to this, the authors may consider ways to make the study generalizable to survivors of childhood, adolescent and young adult cancers living in other countries.

- Yes, we agree it would be useful to make it clear to readers that we are focusing on England and have added this to the title, as suggested.

- By recruiting a diverse range of survivors (e.g., in terms of clinical and socio-demographic characteristics) for the Phase 2 interviews this will benefit the study by enabling us to access a wide range of views and experiences and a comprehensive understanding of smoking in CAYA cancer survivors. 

- These findings may indeed be considered transferrable and applicable to other CAYA survivors (with similar characteristics as those included in our study). Our findings will provide useful insights into the views and experiences of not just the survivors we interview, but of those in the UK and beyond. By presenting the findings of the Phase 2 interviews to survivors in the Phase 3 workshops, we will be able to see how well they relate to the views and experiences reported by other survivors.

It is helpful to more clearly distinguish between studies, recommendations and guidelines that have been produced for England/U.K. and those produced for other countries. This is particularly important where studies address topics that are strongly influenced by policies and healthcare systems that differ across countries.

- We now state that the referenced clinical practice guidelines for survivors are for the UK, US and also international guidelines (Introduction; line 58): Clinical practice guidelines for the follow-up care of CAYA cancer survivors from the UK and US, as well as recent harmonized international guidelines, state that survivors should be advised on both tobacco smoking and smoking cessation.

- Throughout the introduction we have tried to make it clearer what countries the studies concerning health risks in survivors and also smoking rates in survivors have originated from. 

- In the discussion we have highlighted that public health recommendations concerning vaping differ between countries (Discussion; line 373: Contrary to the public health recommendations of other countries (e.g., Australia, US), vaping is promoted as a much safer alternative to smoking in the UK.

The study’s focus is on addressing cessation at the individual/clinical level is useful and appropriate, especially because this population typically has more contact with the health care system than same age peers. However, it is worth noting that population-level interventions (e.g., significant increases in tobacco taxes, smoke-free air laws, anti-tobacco counter marketing) have been shown to promote smoking cessation among the general population of smokers and are also likely to promote cessation in this specific population as well. 

- We agree that population-level interventions are also important for smoking cessation and there are many factors in a person’s wider environment which influences their choice to smoke and their ability to quit. We hope to explore the survivors’ views on these initiatives in the interviews for both smoking and vaping, for instance whether cost influences their smoking/vaping behaviours/desire to quit, where they smoke as smoking is banned in public places and workplaces, and the impact of the wide availability of disposable vapes in the UK.

- Thank you very much for drawing our attention to this document.

I would also draw the authors’ attention to this publication: NCI Tobacco Control Monograph 23, Treating Smoking in Cancer Patients: An Essential Component of Cancer Care. Smoking Cessation among childhood, adolescent and young adult cancer survivors is addressed on pages 92-95. Available here: https://cancercontrol.cancer.gov/brp/tcrb/monographs/monograph-23

- Thank you for drawing our attention to this document too. We were not aware of it, and it looks incredibly useful. 

As I’m sure the authors are aware, smoking cessation typically requires numerous attempts to quit, and tobacco dependence is often viewed as a “chronic condition” until complete and long-term cessation can be achieved. This emphasizes the need for continued vigilance when working with tobacco users, including those who are survivors of childhood, adolescent, and young adult cancers.

- Thank you for highlighting this issue. This issue may be raised by the survivors we interview as we will be asking them about any attempts to quit which will hopefully enable them to share if they have had previously failed attempts and their experiences around this. This is also an issue which we should certainly take into account if developing any smoking cessation intervention. 

Lines 38-40 do not define the comparison/reference group. Suggest do so consistently.

- This text (Introduction; line 43-52) has now been amended to make it clearer what the comparison group was for each study.

Lines 44-46. Suggest explaining that while self-report is considered reliable in the general population, this is not true in cancer survivors and other populations (e.g., pregnant women) who feel significant stigma if they acknowledge tobacco use.

- We have now expanded on this point (Introduction; line 54-57): However, it is worth nothing that these may be underestimates as in clinical groups, where smoking is especially stigmatized (e.g., pregnant women, patients with cardiac disease, cancer patients), patients may be particularly reticent about disclosing their tobacco use.

Lines 78-80. As regards development of smoking cessation interventions targeted to this population, I would encourage the authors to consider both the content of the intervention and the proposed mode of delivery (e.g., in person during clinical visits, telephone quitlines, digital health modalities.)

- Thank you for this comment. In our Phase 3 workshops we hope to touch upon some of the above (e.g., how/where would any intervention be delivered). After this project, should we proceed with developing an intervention for survivors, we will carry out in-depth research and co-design with survivors and stakeholders (including smoking cessation experts) to inform the decisions regarding content and mode. We have edited the text concerning the survivor workshop to make it more clear what issues we expect to discuss and explore (Phase 3; line 323-327): The survivor workshop will explore issues around the acceptability of any intervention to support smoking cessation (e.g., perceived need for an intervention, potential stigma of using intervention), factors which may encourage or prevent CAYA survivors engaging in an intervention (e.g., referral by HCP, low readiness to quit), and initial views on intervention delivery (e.g. where, how and by whom).

Lines 108. Suggest providing the number of survivors who will be included in semi-structured interviews.

- This information has now been added on line 128.

Line 123. As regards the scope of the literature search, suggest the authors consider where it makes sense to restrict to England/UK, and where it is reasonable to expand to other countries. While health care policy and delivery is typically country specific, the experience helping survivors to quit may well be relevant across countries.

- We thank you for drawing our attention to this. We realise that we have not been clear on the limits for our searches. First, we will be carrying out a search of published literature of smoking cessation interventions for CAYA cancer survivors in which we will not be limited in terms of geographical location. The aim of this search is to identify and describe these interventions in the literature. 

- Where we are searching for services specific to England is through the grey literature searches (e.g., Google searches, searching websites) to identify any currently available services for 1) adolescents and/or young people with a medical diagnosis (cancer or another diagnosis) or 2) people of any age with cancer/cancer survivors. The aim of this search is to identify any existing services in England that may be adaptable to the needs of CAYA cancer survivors. 

- In response to your comment, and a comment from reviewer 1 we have amended the text for objective one (line 105) to read: a) To identify and describe the features of tobacco smoking cessation interventions for CAYA cancer survivors which have been published internationally; b) To identify and describe the features of tobacco smoking cessation services which currently exist for i) adolescents and/or young people with a medical diagnosis (cancer or another diagnosis) or ii) people of any age with cancer/cancer survivors in England.

- The text under ‘Overall study design’ (line 126-128) has also been amended to reflect the above and make it the distinction between the two searches. 

- Likewise the abstract has been amended to reflect this (line 10-13): PRISM involves three phases of: 1) an environmental scan using multiple strategies to identify and examine a) smoking cessation interventions for CAYA cancer survivors that are published in the international literature and b) current smoking cessation services in England that may be available to, or tailorable to, CAYA cancer survivors.

Line 207. In addition to variables related to age, gender, diagnosis, and treatment, if efforts will be made to recruit a diverse sample of survivors (e.g., based on socio-economic status, race/ethnicity or other variables) suggest note this.

- Yes, we aim to recruit a diverse sample in terms of ethnicity and socio-economic status, and the other variables we mention as much as we can. We realise we failed to include ethnicity and socio-economic status and have amended this. 

Line 243. I would suggest that the stakeholder interviews examine the survivors’ exposure to second-hand tobacco smoke, at work and in the home. (I am aware that smoking is banned in most workplaces in the UK, but exposure may still occur in these areas.) Exposure to secondhand smoke impedes quitting; conversely, workplace and home smoking bans facilitate quitting. 

- Thank you for the above information – this is really helpful. In the semi-structured interview guide (now included) we have questions which ask about the influence of the people around them (e.g., friends and family) and the environments that they spend their time in. 

Line 328: Use of e-cigarettes has potential implications for smoking cessation efforts. However, it may also be worth noting that not enough is known about the short- and long-term health effects of e-cigarette use, specifically in the high-risk population of cancer survivors. As noted on line 338, e-cigarettes are recommended as a smoking cessation aid in the U.K.; the authors study may wish to consider whether this recommendation is appropriate for childhood, adolescent and young adult cancer survivors.

- We had added that e-cigarettes are only recommended for adult smokers (Discussion; line 360-370): In the UK, e-cigarettes are tightly regulated and recommended as a smoking cessation aid for those aged 18 and above…

- We have also added text to highlight the debate about the safety of vaping (Discussion; line 374-378): Contrary to the public health recommendations of other countries (e.g., Australia, the US), vaping is promoted as a much safer alternative to smoking in the UK. However, little is known about the possible long-term health risks of vaping, and emerging evidence suggests links to increased risk of respiratory and cardiovascular diseases [66], which should be of particular concern to CAYA survivors.

---

## [Editor Report · Decision Letter 1]

9 Feb 2024

Protocol for the Supporting Young Cancer Survivors who Smoke Study (PRISM): Informing the development of a smoking cessation intervention for childhood, adolescent and young adult cancer survivors in England

PONE-D-23-33074R1

Dear Dr. Brown,

We’re pleased to inform you that your manuscript has been judged scientifically suitable for publication and will be formally accepted for publication once it meets all outstanding technical requirements.

Kind regards,

Jesse T. Kaye, PhD

Academic Editor

PLOS ONE

Additional Editor Comments (optional):

I thank both the reviewers and authors for contributing to an excellent revised manuscript. I believe this will be an important and informative study and look forward to reading the results when they are ready. Best of luck to the study team and thank you for submitting this manuscript on this important topic to PLOS One.
---

## [Editor Report · Acceptance letter]

22 Feb 2024

PONE-D-23-33074R1 

PLOS ONE

Dear Dr. Brown, 

I'm pleased to inform you that your manuscript has been deemed suitable for publication in PLOS ONE. Congratulations! Your manuscript is now being handed over to our production team.

Kind regards, 

on behalf of

Dr. Jesse T. Kaye 

Academic Editor

PLOS ONE